# Electronic cigarette use (vaping) and patterns of tobacco cigarette smoking in pregnancy–evidence from a population-based maternity survey in England

Charles Opondo *, Siân Harrison, Fiona Alderdice, Claire Carson, Maria A. Quigley

Nuffield Department of Population Health, NIHR Policy Research Unit in Maternal Health and Care, National Perinatal Epidemiology Unit, University of Oxford, Oxford, United Kingdom

* charles.opondo@npeu.ox.ac.uk

## Abstract

### Objectives

Exposure to tobacco products during pregnancy presents a potential harm to both mother and baby. This study sought to estimate the prevalence of vaping during pregnancy and to explore the factors and outcomes associated with vaping in pregnancy.

### Setting

England.

### Participants

Women who gave birth between 15th and 28th October 2017.

### Methods

A cross-sectional population-based postal survey of maternal and infant health, the National Maternity Survey (NMS) 2018. The prevalence of vaping and patterns of cigarette smoking were estimated, and regression analysis was used to explore associations between maternal characteristics and vaping, and between vaping and birth outcomes.

### Outcome measures

Unweighted and weighted prevalence of vaping with 95% confidence intervals, and unadjusted and adjusted relative risks or difference in means for the association of participant characteristics and secondary outcomes with vaping. Secondary outcome measures were: preterm birth, gestational age at birth, birthweight, and initiation and duration of breastfeeding.

**Data Availability Statement:** The data underlying this study is not publicly available because the scope of the consent obtained from study

participants restricts our ability to share the data on ethical and legal grounds. There are also third-party restrictions by the Office for National Statistics (ONS). Requests to access birth registration data can be submitted to the ONS at https://www.ons. gov.uk/aboutus/whatwedo/statistics/ requestingstatistics/makingarequest; information about the ONS data sharing policy can be found at https://cy.ons.gov.uk/aboutus/ transparencyandgovernance/datastrategy/ datapolicies/onsresearchanddataaccesspolicy. Requests to carry out further analyses on the data from the national maternity surveys can be submitted to the Director of the NPEU at general@npeu.ox.ac.uk. All requests would be subject to the National Perinatal Epidemiology Unit Data Access Policy and may require further regulatory approvals.

**Funding:** This research is funded by the National Institute for Health Research (NIHR) Policy Research Programme, conducted through the Policy Research Unit in Maternal and Neonatal Health and Care, PR-PRU-1217-21202. The views expressed are those of the authors and not necessarily those of the NIHR or the Department of Health and Social Care.

**Competing interests:** All authors have completed the ICMJE uniform disclosure form at http://www. icmje.org/coi_disclosure.pdf. CO, SH, FA, CC, and MQ: no support from any organisation for the submitted work; no financial relationships with any organisations that might have an interest in the submitted work in the previous three years, no other relationships or activities that could appear to have influenced the submitted work. This does not alter our adherence to PLOS ONE policies on sharing data and materials.

## Results

A total of 4,509 women responded to the survey. The prevalence of vaping in pregnancy was 2.8% (95%CI 2.4% to 3.4%). This varied according to the pattern of cigarette smoking in pregnancy: 0.3% in never-smokers; 3.3% in ex-smokers; 7.7% in pregnancy-inspired quitters; 9.5% in temporary quitters; and 17.7% in persistent smokers. Younger women, unmarried women, women with fewer years of formal education, women living with a smoker, and persistent smokers were more likely to vape, although after adjusting for pattern of cigarette smoking and maternal characteristics, persistent smoking was the only risk factor. We did not find any association between vaping and preterm birth, birthweight, or breastfeeding.

## Conclusions

The prevalence of vaping during pregnancy in the NMS 2018 was low overall but much higher in smokers. Smoking was the factor most strongly associated with vaping. Co-occurrence of vaping with persistent smoking has the potential to increase the harms of tobacco exposure in pregnant women and their infants.

## Introduction

An estimated six million people die every year as a result of illnesses related to tobacco use [1]. In pregnancy, tobacco use is associated with harms to both mother and baby, such as placental defects, preeclampsia, stillbirth, preterm birth, low birthweight, sudden infant death, and foetal and infant developmental deficits [2–4]. Tobacco control is therefore a key global health priority. Following a landmark international agreement [5], several countries have taken steps towards controlling tobacco use. Consequently the prevalence of cigarette smoking in the general population, which had been increasing through the 1970s and peaking in the mid 1980s [6], has gradually declined in most countries between 1990 and 2015 [7–10] and is projected to decline further for at least another decade beyond that [11].

The advent of electronic cigarettes–also referred to as e-cigarettes or vaping devices–raises a risk of reversing these important public health gains. Commercially-viable vaping devices began to appear in the mid-2000s, with early models being offered for sale in China in 2004 [12]. Marketed as safer alternatives to conventional smoking [13] and a useful tool to quit cigarette smoking [14], vaping devices became available more widely in the rest of Asia, Europe and the US during the late 2000s. Over the next decade the prevalence of vaping continued to increase globally, followed by reports of morbidity and mortality attributed to vaping [15–18].

In the UK, data from the Office for National Statistics (ONS) show that while the proportion of current cigarette smokers in all four countries of the UK fell from 20.2% in 2011 to 14.7% in 2018 [19], there was a steady increase in the proportion of individuals who reported using vaping devices, from 3.7% in 2014 to 6.3% in 2018. These data also shed some insights into patterns of smoking and vaping across different demographic characteristics. For example, they show that despite the trend of a decline over the period covered, cigarette smoking was most prevalent among individuals aged 25 to 34 years–the peak reproductive age-group– whereas vaping was most prevalent among 35- to 49-year-olds, and that both cigarette smoking and vaping were more prevalent among men than women. Smoking [19] and vaping [20] were also most prevalent in the routine and manual occupation group compared to the managerial group. Across ethnic groups smoking was least prevalent among Chinese respondents

and most prevalent in the mixed ethnic group [19], but no information on patterns of vaping across ethnic groups was available.

For pregnant women who smoke, vaping has been promoted as part of a harm-reduction strategy with the objective of reducing or eliminating tobacco use [14]. However, there remains a dearth of information on the nature and extent of vaping in pregnancy such as would be useful both to health policymakers to inform guidelines on care during pregnancy and to women planning a pregnancy. A recent systematic review explored studies on prevalence, patterns, reasons and health effects of vaping in pregnancy [21]. The review found four studies–all conducted in the US–estimating the prevalence of vaping as between 1.2% and 7.0%. The review also found three studies exploring the effect of vaping on birth outcomes. Two of the studies were based on the same sample, and they explored the association with small-for-gestational age [22, 23] while the third study explored associations with birthweight, gestational age, Apgar scores, and breastfeeding at discharge [24]. All three studies were relatively small: two of them were based on the same US sample of 248 women while the third surveyed 449 women in Ireland.

Our study therefore aimed to estimate the prevalence of vaping during pregnancy, and to explore the factors and outcomes associated with vaping in a population-based sample of women giving birth in England.

## Methods

### Study design

This study was a cross-sectional population-based survey.

### Data sources and acquisition

Data for this study came from the National Maternity Survey (NMS) 2018 [25]. The NMS 2018 was the latest in a series of postal surveys of maternal and infant health and care which are periodically conducted in England by the National Perinatal Epidemiology Unit (NPEU) [26–29].

The Office for National Statistics (ONS) identified, from the birth registration data, a random sample of women who had recently had a baby. Women were eligible for selection if they were aged 16 years or older, were living in England at the time of the birth of their baby, and their baby had been born between 15[th] and 28[th] of October 2017. Women who had had multiple births (e.g. twins) were also included but only asked about their first-born baby. Women were excluded if their baby had died at any time between birth and the time of sampling.

Eligible women were sent a pre-notification card in March 2018 by the ONS to inform them of their selection for the survey and to ask them to look out for the survey questionnaire. The card included contact details and a website address for the study. In April 2018 when the women's babies were around six months old, the ONS sent out survey questionnaires to these women, with return envelopes addressed to the NPEU. A participant information leaflet was enclosed with the questionnaire, which provided detailed information about the study including consent procedures and how the responses from participants would be used. Reminder notifications were sent in May and June 2018 to women who had not responded to initial contacts. Completion and return or submission of a postal or online questionnaire was taken as implicit consent to participate in the study; this approach to obtaining consent was reviewed and approved by the ethics committee. Details about the full methods, content, scope and findings of the NMS are available elsewhere [25].

## Outcomes, explanatory variables, and covariates

The NMS 2018 questionnaire–You & Your Baby–was divided into several sections, starting with 'your pregnancy' and 'your labour and the birth of your baby' and ending with some more general sections including 'your lifestyle'. The lifestyle section included questions on smoking adapted from the Infant Feeding Survey 2010 [30] with additional questions to elicit further details about smoking at different times in the perinatal period. The questions about vaping were adapted from the Opinions and Lifestyle Survey [31] which is published regularly by the ONS. Women were asked about their use of electronic cigarettes or vaping devices since becoming aware of the pregnancy. For vaping, women were asked: "*Did you use an electronic cigarette or vaping device after you found out you were pregnant*?" For smoking, women were asked: "*Did you smoke tobacco cigarettes after you found out you were pregnant*?" Additionally, they were asked how often they smoked during the three months before they became pregnant, during each of the trimesters of their pregnancy, three months, and six months after the birth of their baby (Fig 1). Response categories were 'daily', 'less than daily but at least once a week', 'less than weekly but at least once a month', 'less than monthly' and 'not at all'.

Women's sociodemographic characteristics, including age, country of birth, level of education, area deprivation measured using the index of multiple deprivation (IMD) [32, 33], ethnic group, parity (defined as whether a woman had had a baby previously or not), and whether they were living with their partner at the time of the birth, were available from the NMS data. Information about other circumstances before and during pregnancy, which may have plausibly been related to vaping or smoking status–based on literature–were also available from the NMS data. These pregnancy-related factors included: receiving help and support from partners, relatives, or friends; living with a smoker; whether the pregnancy was planned; and reaction to pregnancy. Birth outcomes were also collected in the survey questionnaire. They included: preterm birth defined as a birth occurring before 37 complete weeks of gestation; gestational age at birth; birthweight; initiation of breastfeeding; and duration of breastfeeding. Observations with birthweights below 500g or above 5,500g or gestational age below 23 weeks or above 43 weeks were excluded from all analyses.

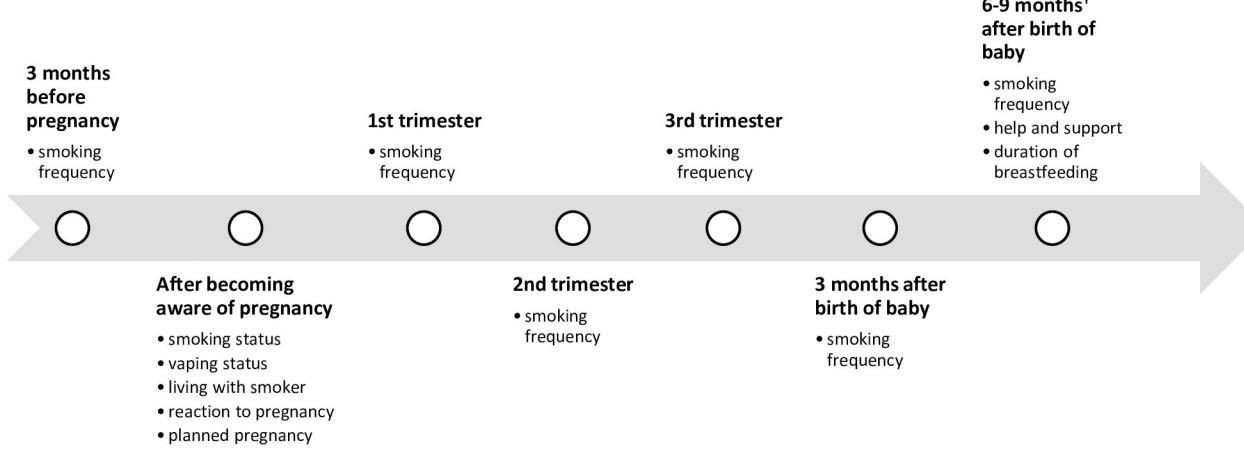

† Majority of women responded 6 to 9 months post-pregnancy even though question asked about *current* smoking at the 6 month postpartum time-point

**Fig 1. Timeline of when the factors explored in this analysis relate to.**

## Sample size

The number of women to be included in the NMS 2018 was based on observations from a pilot study conducted in 2017 [34] which had a response rate of 32%. It was determined that 16,000 women would need to be invited to get responses from 5,120 women if the same response rate was observed in the NMS 2018, or 4,800 women if the response rate was equal to the 30% observed among women who were six months post-partum in the pilot study. A sample of 4,800 women would give at least 90% power to estimate a prevalence of vaping between 2.3% and 4% at the 5% level of statistical significance.

## Analysis

Characteristics of women according to vaping status were summarised using counts and proportions of categorical variables, and means and standard deviations of continuous variables. Where necessary, in categorical variables such as ethnicity where there were small numbers of observations in some groups, larger and more broadly defined groups were created by collapsing similar groups together.

A confirmatory latent class analysis (LCA) was conducted to group women according to their patterns of smoking in pregnancy based on their responses to the question about their frequency of smoking before, during and after their pregnancy. The LCA was constrained to identify five latent classes following a previous study which explored longitudinal patterns of smoking during the pre-conception, pregnancy and postnatal period in the same population [35]. Latent classes were identified using a generalised structural equation model with Poisson distributed self-reported unweighted smoking frequencies. The LCA-predicted smoking classes were then plotted against observed unweighted smoking frequencies to facilitate their interpretation.

The association between women's sociodemographic characteristics and vaping was explored using binomial regression models which estimated the magnitude of association as risk ratios. First, risk ratios for the crude association between each characteristic and vaping were estimated. Next, risk ratios adjusting for the smoking patterns identified in the LCA were estimated, followed by risk ratios adjusting for sociodemographic characteristics for which there was evidence of a crude association with vaping. Lastly, risk ratios further adjusting for smoking patterns identified in the LCA in addition to sociodemographic characteristics which remained associated with vaping after partial adjustment were estimated. The association between vaping and birth outcomes was also explored using the same modelling strategy, with binomial regression for preterm birth, breastfeeding initiation and breastfeeding at 8 weeks, and linear regression for birth weight and gestational age at birth. All analyses were weighted for factors associated with non-response. Weighting ensured that each observation's contribution to overall estimates was appropriately proportional to the representation of their set of characteristics in the sample relative to the national population. Weights were derived using data provided by ONS on responders and non-responders on age, country of birth, IMD, parity, marital status at birth registration, and region of residence. Details of the calculation of weights are described elsewhere [36]. All analyses were conducted using Stata v15.

## Ethical approval

Our institutional Clinical Trials and Research Governance team recommended that we apply for ethical approval through an NHS Research and Ethics Committee (REC). We applied to the NRES Committee for London Bloomsbury, which was holding the next available meeting, and our study was approved (REC reference 18/LO/0271) on 22nd February 2018.

### Patient and public involvement

The research process for the 2018 NMS actively involved patients and the public in the design of the methods, study material, and in the dissemination of survey findings. In addition, the 2018 NMS was based on earlier NMS, which have all relied on extensive patient and public involvement (PPI), and user input from groups such as the National Childbirth Trust.

## Results

### Sample characteristics

A total of 4,509 women completed the NMS 2018, a response rate of 29%. Of these, 88 women did not respond to the question about vaping since becoming aware of the pregnancy, leaving 4,421 women eligible for inclusion in the analysis. The characteristics of the eligible women are summarised in Table 1. The mean age of women was 31 years. Most women were living with a partner, UK-born and of White British ethnicity, were aged 19 or more when they left full-time education, and were having their first baby. Disproportionately more women were from less deprived areas, with just under 16% coming from the most deprived neighbourhoods. We therefore report estimates weighted for the observed response pattern to ensure representativeness of the true distribution of women's characteristics in the population. Due to expectedly high correlation between country of birth and ethnicity (polychoric correlation coefficient -0.80, p-value <0.001), better completeness of country of birth data from the ONS, and small counts in some ethnicity categories, ethnicity was excluded from further analysis.

### Vaping in pregnancy

The weighted prevalence of vaping after becoming aware of pregnancy was 2.8% (95%CI 2.4% to 3.4%). Older women, women living with their partners, and women living in less deprived areas reported lower prevalence of vaping; women born in the UK and women who left full-time education earlier reported higher prevalence of vaping (Table 2). After partial adjustment for all maternal characteristics associated with vaping, only age of woman, country of birth, education and IMD were associated with vaping in pregnancy.

### Cigarette smoking in pregnancy

Of the 4,421 women in the analysis, the pattern of cigarette smoking by 24 women could not be determined due to incomplete data on smoking frequency. Among the remaining 4,397 women, the majority, 2,942 (65.3%) reported never smoking at any time in the period between three months pre-pregnancy and six months after having their baby. Among those who smoked, four groups defined by distinct patterns of smoking frequency were identified in the latent class analysis (Fig 2). One group included women who had previously smoked but mostly reported not smoking at all during this period; these women were described as 'ex-smokers' and comprised 17.6% of the women. The second group included women who had smoked almost daily in the three months before their pregnancy but reduced their frequency of smoking to nearly non-smoking in the first to third trimester, resuming their original frequency of smoking after the birth of their baby; these women were described as 'temporary quitters'; 4.0% of women were in this group. A related, slightly larger group of women (6.2%) maintained their reduced frequency of smoking even after the birth of their baby; they were described as 'pregnancy-inspired quitters'. The fourth group, comprising 6.8% of the sample, continued smoking daily at nearly the same frequency as they did three months before becoming pregnant; these women were described as 'persistent smokers'.

**Table 1. Characteristics of women in the sample.**

| Characteristic | n = 4,421 |
|---|---:|
| Age of woman in years, mean[2] (SD[2]) | 30.8 (5.6) |
| Age when left full-time education, n[1] (%[2]) | |
| 16 years or less | 479 (14.6) |
| 17 to 18 years | 1,033 (26.3) |
| 19 years or more | 2,867 (59.1) |
| Unknown | 42 (0.9) |
| Country of birth, n[1] (%[2]) | |
| Non-UK | 1,009 (29.0) |
| UK | 3,412 (71.0) |
| Ethnicity, n[1] (%[2]) | |
| White British | 3,245 (67.2) |
| White Other | 474 (11.0) |
| Indian, Pakistani, Bangladeshi | 213 (7.2) |
| Black Caribbean, Black African | 88 (3.5) |
| Mixed and other, including Chinese | 262 (7.2) |
| Not stated or missing | 139 (4.0) |
| Index of multiple deprivation (IMD), n[1] (%[2]) | |
| 1 (most deprived) | 693 (27.0) |
| 2 | 852 (22.2) |
| 3 | 929 (18.5) |
| 4 | 981 (17.0) |
| 5 (least deprived) | 966 (15.3) |
| Living with partner, n[1] (%[2]) | |
| No | 313 (15.5) |
| Yes | 4,108 (84.5) |
| Parity, n[1] (%[2]) | |
| Primiparous | 2,284 (51.7) |
| Multiparous | 2,137 (48.3) |
| Smoking cigarettes after awareness of pregnancy, n[1] (%[2]) | |
| No | 4,167 (91.4) |
| Yes | 254 (8.6) |
| Vaping after awareness of pregnancy, n[1] (%[2]) | |
| No | 4,322 (97.2) |
| Yes | 99 (2.8) |

[1]Unweighted

[2]Weighted

## Associations between vaping with cigarette smoking and pregnancy-related factors

There was a trend of increasing prevalence of vaping with increased frequency of smoking (Table 2). In the unadjusted model, compared to ex-smokers, never-smokers were 91% less likely to vape (RR 0.09, 95%CI 0.04 to 0.20). All other groups were more likely to vape than ex-smokers: pregnancy-inspired quitters were 2.40 times more likely to vape (95%CI 1.37 to 4.20), temporary quitters were 2.91 times more likely to vape (95%CI 1.60 to 5.28), and persistent smokers were 5.47 times more likely to vape (95% CI 3.47 to 8.62). The pattern of smoking in pregnancy fully explained all associations between maternal characteristics and vaping. The

**Table 2. Characteristics associated with vaping in pregnancy.**

| Characteristics | Vaping/Total[1] | Proportion vaping (%)[2] | Unadjusted | | Adjusted for smoking pattern only | | Adjusted for sociodemographic characteristics only[3] | | Adjusted for smoking pattern and sociodemographic characteristics[4] | |
|---|---|---|---|---|---|---|---|---|---|---|
| | | | Risk ratio (95% CI)[2] | p-value | Risk ratio (95% CI)[2] | p-value | Risk ratio (95% CI)[2] | p-value | Risk ratio (95% CI)[2] | p-value |
| Age of woman, per year | – | – | 0.91 (0.88 to 0.94) | <0.001 | 0.97 (0.94 to 1.00) | 0.081 | 0.94 (0.91 to 0.98) | 0.001 | 0.98 (0.95 to 1.01) | 0.133 |
| Age when left full-time education | | | | | | | | | | |
| 16 years or less | 25/479 | 5.30 | 3.02 (1.95 to 4.67) | <0.001 | 1.01 (0.65 to 1.57) | 0.936 | 1.83 (1.15 to 2.92) | 0.029 | 0.92 (0.58 to 1.44) | 0.915 |
| 17 to 18 years | 32/1,033 | 4.00 | 2.28 (1.52 to 3.41) | | 1.07 (0.72 to 1.59) | | 1.55 (1.01 to 2.36) | | 0.99 (0.66 to 1.49) | |
| 19 years or more | 41/2,867 | 1.76 | ref. | | ref. | | ref. | | ref. | |
| Country of birth | | | | | | | | | | |
| Non-UK | 11/1,009 | 1.21 | ref. | <0.001 | ref. | | ref. | 0.003 | ref. | 0.352 |
| UK | 88/3,412 | 3.51 | 2.90 (1.71 to 4.92) | | 1.33 (0.79 to 2.23 | 0.282 | 2.32 (1.34 to 4.01) | | 1.28 (0.76 to 2.17) | |
| Index of multiple deprivation (IMD) | | | | | | | | | | |
| 1 (most deprived) | 28/693 | 4.60 | 2.19 (1.23 to 3.90) | 0.001 | 1.10 (0.61 to 1.98) | 0.389 | 1.75 (0.94 to 3.25) | 0.039 | 1.06 (0.59 to 1.92) | 0.648 |
| 2 | 18/852 | 2.32 | 1.11 (0.57 to 2.13) | | 0.74 (0.39 to 1.42) | | 0.99 (0.50 to 1.95) | | 0.76 (0.40 to 1.45) | |
| 3 | 16/929 | 2.20 | 1.05 (0.53 to 2.09) | | 0.77 (0.39 to 1.51) | | 0.95 (0.47 to 1.91) | | 0.78 (0.40 to 1.53) | |
| 4 | 18/981 | 2.09 | 0.99 (0.49 to 2.02) | | 0.80 (0.40 to 1.60) | | 0.91 (0.44 to 1.87) | | 0.80 (0.40 to 1.61) | |
| 5 (least deprived) | 19/966 | 2.10 | ref. | | ref. | | ref. | | ref. | |
| Living with partner | | | | | | | | | | |
| No | 13/313 | 4.83 | ref. | 0.001 | ref. | 0.193 | ref. | 0.976 | – | – |
| Yes | 86/4,108 | 2.47 | 0.51 (0.35 to 0.76) | | 1.30 (0.88 to 1.92) | | 0.99 (0.65 to 1.51) | | – | |
| Parity | | | | | | | | | | |
| Primiparous | 49/2,284 | 2.53 | ref. | 0.292 | – | – | – | – | – | – |
| Multiparous | 50/2,137 | 3.07 | 1.21 (0.85 to 1.73) | | – | | – | | – | |
| Smoking pattern | | | | | | | | | | |
| Never-smoker | 5/2,942 | 0.30 | 0.09 (0.04 to 0.20) | <0.001 | – | – | – | – | 0.09 (0.04 to 0.20) | <0.001 |
| Ex-smoker | 22/880 | 3.33 | ref. | | – | | – | | ref. | |
| Pregnancy-inspired quitter | 21/246 | 7.65 | 2.40 (1.37 to 4.20) | | – | | – | | 2.16 (1.22 to 3.84) | |
| Temporary quitter | 13/137 | 9.48 | 2.91 (1.60 to 5.28) | | – | | – | | 2.51 (1.35 to 4.66) | |
| Persistent smoker | 37/192 | 17.73 | 5.47 (3.47 to 8.62) | | – | | – | | 4.49 (2.71 to 7.45) | |

[1]Unweighted

[2]Weighted

[3]Adjusting for sociodemographic characteristics (age of woman, country of birth, education, IMD quintile and living with partner) for which there is some evidence of unadjusted association with vaping

[4]Adjusting for smoking pattern in pregnancy, in addition to age of woman, country of birth, education and IMD quintile which are still associated with vaping after adjustment for sociodemographic characteristics only

Adjusted estimates are not reported if there is no evidence of association in the unadjusted or adjusted estimate in the previous step of sequential adjustment

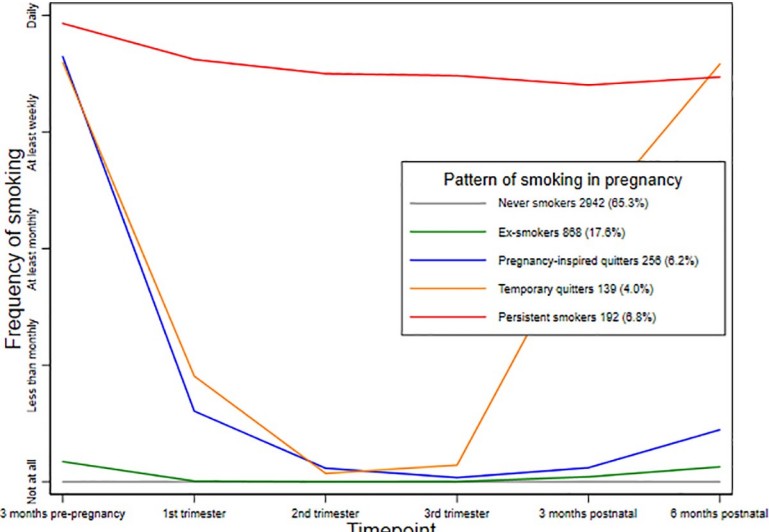

**Fig 2. Latent class analysis of patterns of smoking frequency in pregnancy.**

model adjusting for sociodemographic characteristics only estimated that compared to ex-smokers, never-smokers were 91% less likely to vape (RR 0.09, 95%CI 0.04 to 0.20); pregnancy-inspired quitters were 2.16 times more likely to vape (95%CI 1.22 to 3.84); temporary quitters were 2.51 times more likely to vape (95%CI 1.35 to 4.66); and persistent smokers 4.49 times more likely to vape (95%CI 2.71 to 7.45).

Pregnancy related factors including living with a smoker, unplanned pregnancy, and a neutral or negative reaction to pregnancy were associated with higher prevalence of vaping (Table 3). After adjustment for other sociodemographic factors associated with vaping, only living with a smoker appeared to be associated with increased prevalence of vaping. However, once adjusted for the pattern of smoking in pregnancy, all associations between pregnancy-related factors and vaping were fully explained.

### Associations between vaping and birth outcomes and breastfeeding

Vaping was associated with 12% relative reduction in initiation of breastfeeding (crude RR 0.88, 95%CI 0.79 to 0.97) and 33% relative reduction in breastfeeding for at least eight weeks (crude RR 0.67, 95%CI 0.53 to 0.81), but was not associated with differences in birthweight, gestational age, or the risk of preterm birth (Table 4). After adjustment for age of woman, country of birth, education and IMD only, the pattern of smoking only, or both sets of variables, there was no evidence of association with initiation of breastfeeding or breastfeeding for at least eight weeks.

### Discussion

We used a population-based sample of women who gave birth in England in October 2017 to describe vaping across individual-level characteristics and its associations with birth outcomes and breastfeeding. Recognising the links between vaping and cigarette smoking, we further sought to control for women's patterns of cigarette smoking immediately before, during and after their pregnancy, distinguishing between never-smokers, ex-smokers, pregnancy-inspired quitters, temporary quitters, and persistent smokers.

**Table 3. Pregnancy-related factors associated with vaping.**

| Factors | Vaping/Total[1] | Proportion vaping (%)[2] | Unadjusted | | Adjusted for smoking pattern only | | Adjusted for sociodemographic characteristics only[3] | | Adjusted for smoking pattern and sociodemographic characteristics[4] | |
|---|---|---|---|---|---|---|---|---|---|---|
| | | | Risk ratio (95% CI)[2] | p-value | Risk ratio (95% CI)[2] | p-value | Risk ratio (95% CI)[2] | p-value | Risk ratio (95% CI)[2] | p-value |
| Help and support from others[5] | – | – | 1.02 (0.91 to 1.14) | 0.722 | 0.99 (0.89 to 1.11) | 0.908 | 1.00 (0.89 to 1.12) | 0.959 | – | – |
| Living with someone who smokes | | | | | | | | | | |
| No | 56/3,685 | 1.98 | ref. | | ref. | | ref. | <0.001 | ref. | 0.932 |
| Yes | 41/670 | 6.31 | 3.19 (2.24 to 4.54) | <0.001 | 1.02 (0.71 to 1.47) | 0.917 | 2.34 (1.61 to 3.40) | | 1.02 (0.70 to 1.47) | |
| Planned pregnancy | | | | | | | | | | |
| No | 34/821 | 4.67 | ref. | <0.001 | ref. | | ref. | 0.057 | – | – |
| Yes | 65/3,560 | 2.23 | 0.48 (0.34 to 0.68) | | 1.00 (0.71 to 1.41) | 0.996 | 0.70 (0.48 to 1.01) | | – | |
| Reaction to pregnancy | | | | | | | | | | |
| Positive | 66/3,536 | 2.36 | ref. | 0.002 | ref. | | ref. | 0.167 | – | – |
| Neutral or negative[6] | 30/771 | 4.28 | 1.81 (1.25 to 2.62) | | 1.09 (0.76 to 1.56) | 0.647 | 1.31 (0.89 to 1.93) | | – | |

[1]Unweighted

[2]Weighted

[3]Adjusting for age of woman, country of birth, education and IMD quintile

[4]Adjusting for smoking pattern in pregnancy, in addition to age of woman, country of birth, education and IMD quintile

[5]Measured on a Likert-type scale ranging from 0 which represented no support at all to 6 representing complete support

[6]Out of 55 women who cited a 'negative' reaction only one reported vaping; out of 716 women who cited a 'neutral' reaction, 29 reported vaping

Adjusted estimates are not reported if there is no evidence of association in the unadjusted or adjusted estimate in the previous step of sequential adjustment

**Table 4. Effects of vaping on birth outcomes and breastfeeding.**

| Outcome | Not vaping (N = 4,322) | Vaping (N = 99) | Unadjusted | | Adjusted for smoking pattern only | | Adjusted for sociodemographic characteristics only[3] | | Adjusted for smoking pattern and sociodemographic characteristics[4] | |
|---|---|---|---|---|---|---|---|---|---|---|
| | Mean (SE)[2] | Mean (SE)[2] | Difference (95% CI)[2] | p-value | Difference (95% CI)[2] | p-value | Difference (95% CI)[2] | p-value | Difference (95% CI)[2] | p-value |
| Birthweight[5], grams | 3,365 (9.19) | 3,387 (55.21) | -39.7 (-130.3 to 51.0) | 0.391 | -29.8 (-124.6 to 64.9) | 0.537 | -41.7 (-132.6 to 49.2) | 0.368 | -31.7 (-125.8 to 62.5) | 0.509 |
| Gestational age, weeks | 39.2 (0.03) | 39.0 (0.23) | -0.21 (-0.58 to 0.17) | 0.287 | -0.18 (-0.57 to 0.22) | 0.378 | -0.19 (-0.57 to 0.19) | 0.326 | -0.18 (-0.58 to 0.21) | 0.357 |
| | n[1] (%[2]) | n[1] (%[2]) | Risk ratio (95% CI)[2] | p-value | Risk ratio (95% CI)[2] | p-value | Risk ratio (95% CI)[2] | p-value | Risk ratio (95% CI)[2] | p-value |
| Preterm birth | 303 (7.34) | 8 (9.37) | 1.28 (0.57 to 1.98) | 0.394 | 1.30 (0.54 to 2.05) | 0.390 | 1.25 (0.55 to 1.95) | 0.444 | 1.29 (0.54 to 2.04) | 0.403 |
| Initiation of breastfeeding | 3,847 (85.60) | 79 (75.12) | 0.88 (0.79 to 0.97) | 0.001 | 1.03 (0.97 to 1.08) | 0.379 | 1.00 (0.95 to 1.06) | 0.973 | 1.03 (0.98 to 1.08) | 0.210 |
| Breastfeeding for at least 8 weeks | 2,848 (61.48) | 42 (41.36) | 0.67 (0.53 to 0.81) | <0.001 | 0.99 (0.84 to 1.14) | 0.914 | 0.92 (0.79 to 1.05) | 0.201 | 1.04 (0.91 to 1.17) | 0.571 |

[1]Unweighted

[2]Weighted

[3]Adjusting for age of woman, country of birth, education and IMD quintile

[4]Adjusting for smoking pattern in pregnancy, in addition to age of woman, country of birth, education and IMD quintile

[5]Gestational age-adjusted

We found that 2.8% (95%CI 2.4% to 3.4%) of the women sampled were vaping after they were aware of their pregnancy, with a higher adjusted prevalence of vaping among younger women, women born in the UK, women who left full-time education at a younger age, women living in the most deprived areas, and women who lived with a smoker during their pregnancy. Women's smoking patterns were strongly associated with vaping and ranged from an adjusted prevalence of vaping in never-smokers being a tenth of that in ex-smokers, to an over five-fold relative increase among persistent smokers. Further adjusting the associations between women's sociodemographic characteristics and vaping for smoking patterns fully explained all associations; this could in part be due to commonality between the characteristics of smokers and vapers, in particular the indicators of socioeconomic position such as education, IMD and marital status. Vaping was also associated with lower probability of breastfeeding initiation and lower probability of breastfeeding for at least eight weeks, although these associations were fully explained by maternal characteristics associated with vaping upon adjustment.

The prevalence of vaping estimated in this study was lower than that from a survey of women who were attending hospital antenatal care appointments at 8 to 24 weeks into their pregnancy, which was conducted in England and Scotland between June and November 2017 [37]. Like our study, the antenatal survey found that most of the women who vaped also smoked, however they reported a higher prevalence of current vaping than we observed: 4.8% (95%CI 4.1 to 5.6), of which 1.3% (95%CI 1.0 to 1.8) were exclusive vapers and 3.5% (95%CI 2.9 to 4.2) were dual cigarette smokers and vapers. The difference in methodological approaches may explain some of the observed discrepancy in the prevalence estimates: the antenatal survey was based on screening for a study of attitudes to vaping among women at selected hospital sites, as such it is less likely to be representative of the national population and more prone to selection bias than the NMS 2018 which identified and invited all eligible women to participate. For example, non-smoking or non-vaping women may have been more likely to take part in the NMS 2018 because it was about health in general, but less likely to participate in a survey about smoking.

Our findings are more similar to those from a US survey of women aged 18 to 44 years conducted between 2014 and 2017 which found a 3.6% prevalence of vaping in pregnancy (versus 2.8% in this sample) and an 8% prevalence of cigarette smoking (compared to the 6.8% persistent smokers in this sample) [38]. While the US study observed that the prevalence of vaping among pregnant women was 38.9% in current cigarette smokers, 1.3% in ex-smokers, and 0.3% in never-smokers, our study found a 34.9% prevalence among women who smoked at any point during pregnancy (including quitters), 3.3% among ex-smokers and 0.3% among never smokers. However, unlike the US study which compared vaping and smoking in both pregnant and non-pregnant women, our study was limited to pregnant women only. Importantly, the US study observed similar prevalence of vaping in both pregnant and non-pregnant women but lower prevalence of cigarette smoking in pregnant women. Nevertheless other US studies have observed a similar pattern of higher prevalence of vaping in pregnancy among current or recent cigarette smokers [39–41], and a systematic review found a range of prevalence estimates ranging between 1.2% and 7.0% across studies [21].

In the general population of adults in England, vaping is more common among cigarette smokers than non- or ex-smokers [42] and this pattern is likely to be maintained among pregnant women. The higher prevalence of vaping among women who continued smoking throughout their pregnancy and afterwards compared to women who quit either temporarily or beyond the end of pregnancy could indicate the use of vaping in addition to, rather than as a substitute for, cigarette smoking. It may also indicate women who were more frequent smokers attempted to use vaping to quit smoking but were unsuccessful. This may have adverse implications on the effectiveness of vaping as a smoking cessation aid for pregnant women;

indeed a systematic review exploring smoking cessation among electronic cigarette users compared to non-users found lower odds of success of quitting cigarettes among vapers, whether or not they took up vaping with an intention to quit smoking [43]. A more recent review found higher rates of quitting among users of nicotine-containing vaping devices compared to individuals using non-nicotine vaping devices (moderate-certainty evidence), nicotine replacement therapy (moderate-certainty evidence), or behavioural support or no support (low-certainty evidence) [44]. Another study found evidence of greater likelihood of transitioning to vaping in addition to–rather than as substitute for–cigarette smoking among pregnant women [45]. Nevertheless, successfully replacing smoking with vaping may be beneficial in pregnancy because vaping is thought to be associated with a lower risk of harm compared to cigarette smoking [46, 47].

Women who reported vaping were less likely to initiate breastfeeding, although the effect of vaping on breastfeeding was reduced after accounting for country of birth and IMD, highlighting the strong social patterning of vaping by pregnant women. Vaping women who initiated breastfeeding were more likely to stop breastfeeding sooner than those who did not vape. Both of these findings are consistent with established evidence of reduced prevalence and duration of breastfeeding among women who smoke cigarettes [48, 49]. These effects have been attributed to reduced milk production and shorter lactation periods in women exposed to tobacco [50, 51]. There is also evidence of an adverse effect on milk production and lactation among women exposed to second-hand smoke [52] suggesting that the higher prevalence of vaping among women living with smokers could compound the negative consequences of vaping on breastfeeding. The home environment may also contribute to the lower rate of initiation of breastfeeding among women who vaped, as implied by the finding that women living with a smoker were more likely to vape. This is consistent with evidence that women living with a smoker were more likely to continue smoking during their pregnancy and to relapse after pregnancy if they quit during pregnancy [53].

Although previous studies have highlighted evidence of increased risk of adverse birth outcomes such as small-for-gestational-age, low birthweight, and preterm birth, with exposure to tobacco through vaping [54–56], we did not observe any associations between vaping and preterm birth, gestation, or birthweight. This could be attributed to various factors: for example, our study had a comparatively small number of women reporting vaping, resulting in relatively lower power to identify associations. Another potential limitation was that exposures and outcomes were measured retrospectively through self-report. Self-reported outcomes could be subject to recall bias, and self-reported measures of smoking and vaping in pregnancy could be subject to social desirability bias which could result in an underestimate of the true prevalence and frequency of vaping and smoking; indeed a lower estimate of the prevalence of smoking at the time of giving birth was observed in the NMS 2018 (7%) compared to a contemporaneous survey by the Department of Health and Social Care (10.8%) [25] in which data on prevalence was collected by the midwife. Retrospective measurement of events occurring up to 21 months in the past could be subject to recall bias. As with other observational studies, there is a limit to the extent to which causal inferences can be drawn from this study, especially due to the possibility of residual confounding by other factors not measured or adjusted for. The lack of detailed data on vaping at different time-points before, during and after pregnancy, similar to the smoking data, or on frequency, quantity, type of device or level of nicotine in vaping products used, was another limitation. The NMS 2018 pilot study [34] had included questions on vaping at different time-points, but the prevalence of vaping at many of the time-points was so low that the detailed questions were deemed not to be useful, and the format of the question was revised for the main study to assess prevalence of vaping of any frequency after awareness of pregnancy. Data on the number of cigarettes smoked per day was also not collected. Lastly,

the response rate to the NMS 2018 was low, although this was consistent with a secular trend of declining response rates in other recent population based surveys in England [57]. Nevertheless the use of appropriately-derived survey weights [58] is likely to have resulted in valid estimates even with these low response rates.

Despite these limitations, this study featured a large enough sample of women to robustly estimate the prevalence of vaping and identify its associations with other factors. The well-established format of the NMS also helped to obtain a representative sample of women who had recently had a baby. Availability of additional individual characteristics of all eligible women from the ONS also aided the derivation of suitable sampling weights to further improve our estimates.

## Conclusions

The prevalence of vaping among pregnant women in England is low. Nevertheless, vaping appears to be more common among women who smoke persistently during their pregnancy than among those who quit or do not smoke. If the prevalence of vaping continues to increase as suggested by current trends and if vaping in pregnancy continues to co-occur with persistent cigarette smoking rather than cessation, then there is potential for increased prevalence of adverse outcomes of pregnancy and birth. This study recommends the development of alternative smoking cessation or smoking reduction tools for particular use in pregnancy which limit harms to mother and baby.

## Acknowledgments

Most thanks are due to the many women who participated in the surveys and to the women who provided input into the development of the NMS. Staff at the Office for National Statistics drew the samples and managed the mailings but bear no responsibility for analysis or interpretation of the data.

## Author Contributions

**Conceptualization:** Siân Harrison, Fiona Alderdice, Claire Carson, Maria A. Quigley.

**Data curation:** Siân Harrison, Fiona Alderdice, Maria A. Quigley.

**Formal analysis:** Charles Opondo.

**Funding acquisition:** Fiona Alderdice, Maria A. Quigley.

**Investigation:** Charles Opondo, Siân Harrison, Fiona Alderdice, Claire Carson, Maria A. Quigley.

**Methodology:** Charles Opondo, Siân Harrison, Fiona Alderdice, Claire Carson, Maria A. Quigley.

**Supervision:** Fiona Alderdice, Maria A. Quigley.

**Writing – original draft:** Charles Opondo.

**Writing – review & editing:** Charles Opondo, Siân Harrison, Fiona Alderdice, Claire Carson, Maria A. Quigley.

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
