## [Decision Letter · Decision Letter 0]

15 Mar 2021

PONE-D-21-04030

Electronic cigarette use (vaping) and patterns of tobacco cigarette smoking in pregnancy – evidence from a population-based maternity survey in England

PLOS ONE

Dear Dr. Opondo,

Thank you for submitting your manuscript to PLOS ONE. After careful consideration, we feel that it has merit but does not fully meet PLOS ONE’s publication criteria as it currently stands. Therefore, we invite you to submit a revised version of the manuscript that addresses the points raised during the review process.

We look forward to receiving your revised manuscript.

Kind regards,

Frank T. Spradley

Academic Editor

PLOS ONE

2. Please provide a sample size and power calculation in the Methods, or discuss the reasons for not performing one before study initiation.

3. Please state whether the questionnaire provided participants with information about the study and how their responses would be used. Please also state whether participants were required to provide written informed consent to be included in the study.

4. In the ethics statement in the manuscript Methods, please explain why ethics approval was obtained from a review board that is not associated with your institution.

Reviewers' comments:

Reviewer's Responses to Questions

**Comments to the Author**

1. Is the manuscript technically sound, and do the data support the conclusions?

Reviewer #1: Yes

Reviewer #2: Yes

Reviewer #3: Yes

2. Has the statistical analysis been performed appropriately and rigorously? 

Reviewer #1: Yes

Reviewer #2: I Don't Know

Reviewer #3: Yes

3. Have the authors made all data underlying the findings in their manuscript fully available?

Reviewer #1: Yes

Reviewer #2: Yes

Reviewer #3: No

4. Is the manuscript presented in an intelligible fashion and written in standard English?

Reviewer #1: Yes

Reviewer #2: No

Reviewer #3: Yes

5. Review Comments to the Author

Reviewer #1: This study involved a cross-sectional population-based postal survey (N=4421) of maternal and infant health and examined the prevalence of vaping and associations between vaping and maternal characteristics. Vaping has the potential for both maternal and infant harm as well as for aiding smoking cessation; therefore, it is important to know about both the prevalence of vaping and factors associated with vaping. This is a rigorously conducted study and is likely to have a more representative sample than the previous similar survey by Bowker and colleagues (ref 33). The conclusions seem well justified and I co nsider this a valubale contribution to the literature. I only have a couple of suggestions relating to the referencing of related literature.

I think far too much of the introduction is spent discussing smoking and vaping in the general population and the authors need to place the study more in the context of the growing literature on vaping in pregnancy. This very recent review might be helpful:

Robert Calder, PhD, Eleanor Gant, MSc, Linda Bauld, PhD, Ann McNeill, PhD, Debbie Robson, PhD, Leonie S Brose, PhD, Vaping in pregnancy: A systematic review Nicotine & Tobacco Research

https://doi.org/10.1093/ntr/ntab017

Published: 04 February 2021

When the authors mention “a systematic review exploring smoking cessation among electronic cigarette users found lower odds of success of quitting cigarettes among vapers, whether or not they took up vaping with an intention to quit smoking36.”, I suggest they offer a more balanced perspective by also referring to the findings of the recent Cochrane review on vaping for smoking cessation:

https://www.cochranelibrary.com/cdsr/doi/10.1002/14651858.CD010216.pub4/full

Reviewer #2: The findings from the LCA are not reported, other than saying classes were formed. What model fit indicies were used to determine the classes, and where were the findings from the LCA reported? Also, the paper includes several typographical errors and needs revisions throughout (abstract to discussion) to address these.

Reviewer #3: This paper examined the prevalence of e-cigarette use (vaping) during pregnancy among a cohort of women in England and the associations between vaping and birth outcomes and breastfeeding. This is an important area as there is limited known about vaping in this population as well as the impact on infant health. While the authors have produced a compelling paper, I would like to raise two methodological points.

First, I would like to comment upon how the authors have structured adjusting in their models in Tables 2-4. There is a substantial body of research establishing the associations between smoking cigarettes with socio-demographic characteristics and birth outcomes. Rather than adjusting for smoking patterns in the ‘further adjusted model’, I would suggest they consider only adjusting for cigarette use before they adjust for the additional socio-demographic characteristics. This would then isolate the effect of vaping independent of smoking on these characteristics. It is not currently clear why adjusting for smoking is left to the ‘further adjusted model’ as smoking is likely the main driver for many of these associations.

Second, the authors commented on page 7 that because the variables country of birth and ethnicity were ‘correlated’ then they have chosen to only include country of birth in their model. It would helpful if the authors produced a table or describe in text how much overlap there is between the two variables. As smoking is known to vary by ethnicity, it seems the authors are missing the opportunity to look at this important risk factor. One suggestion is to run sensitivity analyses and substitute ‘ethnicity’ for ‘country of birth’ to see whether and how the associations change between these two approaches.

Abstract

• The authors should include the N of the survey

• The results report that there was no association between vaping and breastfeeding; however, the conclusion states that ‘reduced breastfeeding among women…’ While this is more clearly described in the results (in the paper), this sounds contradictory in the abstract

Introduction

• While the authors introduce 1 study on the prevalence of e-cigarette use during pregnancy in the US in the discussion, I think the authors should include a paragraph describing the limited research in this area related to the prevalence and associations with birth outcomes. The authors should also consider citing Calder et al. Nicotine & Tobacco Research 2021: Vaping in pregnancy: A systematic review

Methods

• A limitation of the collection of vaping and smoking data is how the questions were asked. In the limitations section (in discussion), the authors should note that there is no information on the quantity of consumption (i.e. how many times vaped during that time period), type of device used, and level of nicotine contained in the vaping products. Also, was the number of cigarettes smoked per day collected? From this description it does not appear so, but this would also be a limitation that should be acknowledged.

• The socio-demographic characteristics listed in the paragraph do not match all that were included in Table 1, including education and index of multiple deprivation. The last sentence comments that the birth outcomes and breastfeeding were collected in the survey. From this statement it is not clear if this information was also collected by self-report. If this is the case, then that should be noted as birth weight and gestational age are not likely accurately recalled and should be noted in the limitations section.

• The authors should include the definition of preterm birth.

• It would also be helpful to know whether the birth outcome data were cleaned, including excluding implausible birth weights for gestational age or any other exclusions of the data such as age restrictions or how missing data were handled.

• It would be helpful if the authors could clarify why they chose binomial regression models rather than logistic regression models for the dichotomous outcomes.

• To clarify, the authors mention that weights were included to adjust for non-response and thus make the results nationally-representative, correct? Or simply to adjust for non-response.

Results

• The first paragraph states ‘married’ but the table indicates the definition was ‘living with partner’

• On page 8, the authors note that the pattern of smoking could not be determined in 24 women but Table 1 still includes these women. It seems they should be excluded from the analytic sample as they will not be included in the adjusted regression models.

• On page 9, it is not clear why ‘ex-smokers’ is the baseline group for the smoking pattern variable (Table 2). Shouldn’t everyone be compared to ‘never-smokers’?

Discussion

• On page 13, the authors only describe 1 study from the US that examined the prevalence of prenatal e-cigarette use and there are multiple studies published on this topic. While a comprehensive review is not needed, it would be helpful to describe existing studies more fully.

• The discussion describes the association between vaping and smoking and I think adjusting for smoking first would contribute to the points raised, particularly in relation to the birth outcomes and breastfeeding.

• As noted previously, the limitations section should be expanded to comment upon how the birth outcomes were collected and the information that wasn’t collected in the questions related to vaping and smoking noted previously.

6. PLOS authors have the option to publish the peer review history of their article (what does this mean?). If published, this will include your full peer review and any attached files.

Reviewer #1: **Yes: **Professor Michael Ussher

Reviewer #2: No

Reviewer #3: No

---

## [Author Response · Author response to Decision Letter 0]

28 Apr 2021

Detailed responses to reviewers' comments are included at the end of this document.

---

## [Decision Letter · Decision Letter 1]

24 May 2021

Electronic cigarette use (vaping) and patterns of tobacco cigarette smoking in pregnancy – evidence from a population-based maternity survey in England

PONE-D-21-04030R1

Dear Dr. Opondo,

We’re pleased to inform you that your manuscript has been judged scientifically suitable for publication and will be formally accepted for publication once it meets all outstanding technical requirements.

Kind regards,

Frank T. Spradley

Academic Editor

PLOS ONE

Reviewers' comments:

Reviewer's Responses to Questions

**Comments to the Author**

1. If the authors have adequately addressed your comments raised in a previous round of review and you feel that this manuscript is now acceptable for publication, you may indicate that here to bypass the “Comments to the Author” section, enter your conflict of interest statement in the “Confidential to Editor” section, and submit your "Accept" recommendation.

Reviewer #1: All comments have been addressed

Reviewer #2: All comments have been addressed

Reviewer #3: All comments have been addressed

2. Is the manuscript technically sound, and do the data support the conclusions?

Reviewer #1: Yes

Reviewer #2: Yes

Reviewer #3: Yes

3. Has the statistical analysis been performed appropriately and rigorously? 

Reviewer #1: Yes

Reviewer #2: Yes

Reviewer #3: Yes

4. Have the authors made all data underlying the findings in their manuscript fully available?

Reviewer #1: Yes

Reviewer #2: Yes

Reviewer #3: Yes

5. Is the manuscript presented in an intelligible fashion and written in standard English?

Reviewer #1: Yes

Reviewer #2: Yes

Reviewer #3: Yes

6. Review Comments to the Author

Reviewer #1: The authors have done a good job of responding to the reviewer's comments and I have no further request for revisions.

Reviewer #2: The authors have addressed most recommended comments. The introducton could benefit from narrowing the focus to the target population fasters, but overall, the paper is improved from the original submission.

Reviewer #3: The authors have adequately addressed the points raised by the reviewer.

7. PLOS authors have the option to publish the peer review history of their article (what does this mean?). If published, this will include your full peer review and any attached files.

Reviewer #1: **Yes: **Professor Michael Ussher

Reviewer #2: No

Reviewer #3: No

---

## [Editor Report · Acceptance letter]

26 May 2021

PONE-D-21-04030R1 

Electronic cigarette use (vaping) and patterns of tobacco cigarette smoking in pregnancy – evidence from a population-based maternity survey in England 

Dear Dr. Opondo:

I'm pleased to inform you that your manuscript has been deemed suitable for publication in PLOS ONE. Congratulations! Your manuscript is now with our production department. 

Kind regards, 

on behalf of

Dr. Frank T. Spradley 

Academic Editor

PLOS ONE